# Skip-Thought Vectors

**Ryan Kiros** [1], **Yukun Zhu** [1], **Ruslan Salakhutdinov** [1,2], **Richard S. Zemel** [1,2]
**Antonio Torralba** [3], **Raquel Urtasun** [1], **Sanja Fidler** [1]
University of Toronto [1]
Canadian Institute for Advanced Research [2]
Massachusetts Institute of Technology [3]

## Abstract

We describe an approach for unsupervised learning of a generic, distributed sentence encoder. Using the continuity of text from books, we train an encoder-decoder model that tries to reconstruct the surrounding sentences of an encoded passage. Sentences that share semantic and syntactic properties are thus mapped to similar vector representations. We next introduce a simple vocabulary expansion method to encode words that were not seen as part of training, allowing us to expand our vocabulary to a million words. After training our model, we extract and evaluate our vectors with linear models on 8 tasks: semantic relatedness, paraphrase detection, image-sentence ranking, question-type classification and 4 benchmark sentiment and subjectivity datasets. The end result is an off-the-shelf encoder that can produce highly generic sentence representations that are robust and perform well in practice.

## 1 Introduction

Developing learning algorithms for distributed compositional semantics of words has been a long-standing open problem at the intersection of language understanding and machine learning. In recent years, several approaches have been developed for learning composition operators that map word vectors to sentence vectors including recursive networks [1], recurrent networks [2], convolutional networks [3, 4] and recursive-convolutional methods [5, 6] among others. All of these methods produce sentence representations that are passed to a supervised task and depend on a class label in order to backpropagate through the composition weights. Consequently, these methods learn high-quality sentence representations but are tuned only for their respective task. The paragraph vector of [7] is an alternative to the above models in that it can learn unsupervised sentence representations by introducing a distributed sentence indicator as part of a neural language model. The downside is at test time, inference needs to be performed to compute a new vector.

In this paper we abstract away from the composition methods themselves and consider an alternative loss function that can be applied with any composition operator. We consider the following question: is there a task and a corresponding loss that will allow us to learn highly generic sentence representations? We give evidence for this by proposing a model for learning high-quality sentence vectors without a particular supervised task in mind. Using word vector learning as inspiration, we propose an objective function that abstracts the skip-gram model of [8] to the sentence level. That is, instead of using a word to predict its surrounding context, we instead encode a sentence to predict the sentences around it. Thus, any composition operator can be substituted as a sentence encoder and only the objective function becomes modified. Figure 1 illustrates the model. We call our model **skip-thoughts** and vectors induced by our model are called **skip-thought vectors**.

Our model depends on having a training corpus of contiguous text. We chose to use a large collection of novels, namely the BookCorpus dataset [9] for training our models. These are free books written by yet unpublished authors. The dataset has books in 16 different genres, e.g., *Romance* (2,865 books), *Fantasy* (1,479), *Science fiction* (786), *Teen* (430), etc. Table 1 highlights the summary statistics of the book corpus. Along with narratives, books contain dialogue, emotion and a wide range of interaction between characters. Furthermore, with a large enough collection the training set is not biased towards any particular domain or application. Table 2 shows nearest neighbours

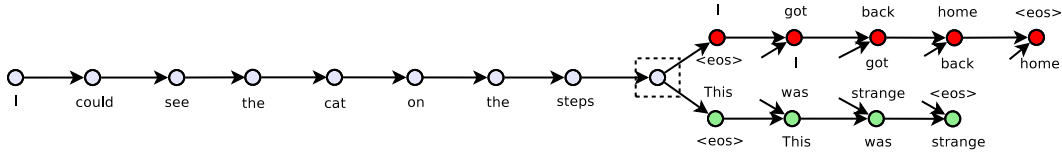

Figure 1: The skip-thoughts model. Given a tuple $(s_{i-1}, s_i, s_{i+1})$ of contiguous sentences, with $s_i$ the $i$-th sentence of a book, the sentence $s_i$ is encoded and tries to reconstruct the previous sentence $s_{i-1}$ and next sentence $s_{i+1}$. In this example, the input is the sentence triplet *I got back home. I could see the cat on the steps. This was strange.* Unattached arrows are connected to the encoder output. Colors indicate which components share parameters. $\langle eos \rangle$ is the end of sentence token.

| # of books | # of sentences | # of words | # of unique words | mean # of words per sentence |
|---|---|---|---|---|
| 11,038 | 74,004,228 | 984,846,357 | 1,316,420 | 13 |

Table 1: Summary statistics of the **BookCorpus dataset** [9]. We use this corpus to training our model.

of sentences from a model trained on the BookCorpus dataset. These results show that skip-thought vectors learn to accurately capture semantics and syntax of the sentences they encode.

We evaluate our vectors in a newly proposed setting: after learning skip-thoughts, freeze the model and use the encoder as a generic feature extractor for arbitrary tasks. In our experiments we consider 8 tasks: semantic-relatedness, paraphrase detection, image-sentence ranking and 5 standard classification benchmarks. In these experiments, we extract skip-thought vectors and train linear models to evaluate the representations directly, without any additional fine-tuning. As it turns out, skip-thoughts yield generic representations that perform robustly across all tasks considered.

One difficulty that arises with such an experimental setup is being able to construct a large enough word vocabulary to encode arbitrary sentences. For example, a sentence from a Wikipedia article might contain nouns that are highly unlikely to appear in our book vocabulary. We solve this problem by learning a mapping that transfers word representations from one model to another. Using pre-trained word2vec representations learned with a continuous bag-of-words model [8], we learn a linear mapping from a word in word2vec space to a word in the encoder's vocabulary space. The mapping is learned using all words that are shared between vocabularies. After training, any word that appears in word2vec can then get a vector in the encoder word embedding space.

## 2 Approach

### 2.1 Inducing skip-thought vectors

We treat skip-thoughts in the framework of encoder-decoder models [1]. That is, an encoder maps words to a sentence vector and a decoder is used to generate the surrounding sentences. Encoder-decoder models have gained a lot of traction for neural machine translation. In this setting, an encoder is used to map e.g. an English sentence into a vector. The decoder then conditions on this vector to generate a translation for the source English sentence. Several choices of encoder-decoder pairs have been explored, including ConvNet-RNN [10], RNN-RNN [11] and LSTM-LSTM [12]. The source sentence representation can also dynamically change through the use of an attention mechanism [13] to take into account only the relevant words for translation at any given time. In our model, we use an RNN encoder with GRU [14] activations and an RNN decoder with a conditional GRU. This model combination is nearly identical to the RNN encoder-decoder of [11] used in neural machine translation. GRU has been shown to perform as well as LSTM [2] on sequence modelling tasks [14] while being conceptually simpler. GRU units have only 2 gates and do not require the use of a cell. While we use RNNs for our model, any encoder and decoder can be used so long as we can backpropagate through it.

Assume we are given a sentence tuple $(s_{i-1}, s_i, s_{i+1})$. Let $w_i^t$ denote the $t$-th word for sentence $s_i$ and let $\mathbf{x}_i^t$ denote its word embedding. We describe the model in three parts: the encoder, decoder and objective function.

**Encoder.** Let $w_i^1, \ldots, w_i^N$ be the words in sentence $s_i$ where $N$ is the number of words in the sentence. At each time step, the encoder produces a hidden state $\mathbf{h}_i^t$ which can be interpreted as the representation of the sequence $w_i^1, \ldots, w_i^t$. The hidden state $\mathbf{h}_i^N$ thus represents the full sentence.

| Query and nearest sentence |
| --- |
| he ran his hand inside his coat , double-checking that the unopened letter was still there .<br>he slipped his hand between his coat and his shirt , where the folded copies lay in a brown envelope . |
| im sure youll have a glamorous evening , she said , giving an exaggerated wink .<br>im really glad you came to the party tonight , he said , turning to her . |
| although she could tell he had n't been too invested in any of their other chitchat , he seemed genuinely curious about this .<br>although he had n't been following her career with a microscope , he 'd definitely taken notice of her appearances . |
| an annoying buzz started to ring in my ears , becoming louder and louder as my vision began to swim .<br>a weighty pressure landed on my lungs and my vision blurred at the edges , threatening my consciousness altogether . |
| if he had a weapon , he could maybe take out their last imp , and then beat up errol and vanessa .<br>if he could ram them from behind , send them sailing over the far side of the levee , he had a chance of stopping them . |
| then , with a stroke of luck , they saw the pair head together towards the portaloos .<br>then , from out back of the house , they heard a horse scream probably in answer to a pair of sharp spurs digging deep into its flanks . |
| " i 'll take care of it , " goodman said , taking the phonebook .<br>" i 'll do that , " julia said , coming in . |
| he finished rolling up scrolls and , placing them to one side , began the more urgent task of finding ale and tankards .<br>he righted the table , set the candle on a piece of broken plate , and reached for his flint , steel , and tinder . |

Table 2: In each example, the first sentence is a query while the second sentence is its nearest neighbour. Nearest neighbours were scored by cosine similarity from a random sample of 500,000 sentences from our corpus.

To encode a sentence, we iterate the following sequence of equations (dropping the subscript $i$):

$$
\mathbf{r}^t = \sigma(\mathbf{W}_r \mathbf{x}^t + \mathbf{U}_r \mathbf{h}^{t-1}) \tag{1}
$$

$$
\mathbf{z}^t = \sigma(\mathbf{W}_z \mathbf{x}^t + \mathbf{U}_z \mathbf{h}^{t-1}) \tag{2}
$$

$$
\bar{\mathbf{h}}^t = \tanh(\mathbf{W}\mathbf{x}^t + \mathbf{U}(\mathbf{r}^t \odot \mathbf{h}^{t-1})) \tag{3}
$$

$$
\mathbf{h}^t = (1 - \mathbf{z}^t) \odot \mathbf{h}^{t-1} + \mathbf{z}^t \odot \bar{\mathbf{h}}^t \tag{4}
$$

where $\bar{\mathbf{h}}^t$ is the proposed state update at time $t$, $\mathbf{z}^t$ is the update gate, $\mathbf{r}_t$ is the reset gate ($\odot$) denotes a component-wise product. Both update gates takes values between zero and one.

**Decoder.** The decoder is a neural language model which conditions on the encoder output $\mathbf{h}_i$. The computation is similar to that of the encoder except we introduce matrices $\mathbf{C}_z$, $\mathbf{C}_r$ and $\mathbf{C}$ that are used to bias the update gate, reset gate and hidden state computation by the sentence vector. One decoder is used for the next sentence $s_{i+1}$ while a second decoder is used for the previous sentence $s_{i-1}$. Separate parameters are used for each decoder with the exception of the vocabulary matrix $\mathbf{V}$, which is the weight matrix connecting the decoder's hidden state for computing a distribution over words. In what follows we describe the decoder for the next sentence $s_{i+1}$ although an analogous computation is used for the previous sentence $s_{i-1}$. Let $\mathbf{h}^t_{i+1}$ denote the hidden state of the decoder at time $t$. Decoding involves iterating through the following sequence of equations (dropping the subscript $i+1$):

$$
\mathbf{r}^t = \sigma(\mathbf{W}^d_r \mathbf{x}^{t-1} + \mathbf{U}^d_r \mathbf{h}^{t-1} + \mathbf{C}_r \mathbf{h}_i) \tag{5}
$$

$$
\mathbf{z}^t = \sigma(\mathbf{W}^d_z \mathbf{x}^{t-1} + \mathbf{U}^d_z \mathbf{h}^{t-1} + \mathbf{C}_z \mathbf{h}_i) \tag{6}
$$

$$
\bar{\mathbf{h}}^t = \tanh(\mathbf{W}^d \mathbf{x}^{t-1} + \mathbf{U}^d(\mathbf{r}^t \odot \mathbf{h}^{t-1}) + \mathbf{C}\mathbf{h}_i) \tag{7}
$$

$$
\mathbf{h}^t_{i+1} = (1 - \mathbf{z}^t) \odot \mathbf{h}^{t-1} + \mathbf{z}^t \odot \bar{\mathbf{h}}^t \tag{8}
$$

Given $\mathbf{h}^t_{i+1}$, the probability of word $w^t_{i+1}$ given the previous $t-1$ words and the encoder vector is

$$
P(w^t_{i+1} | w^{<t}_{i+1}, \mathbf{h}_i) \propto \exp(\mathbf{v}_{w^t_{i+1}} \mathbf{h}^t_{i+1}) \tag{9}
$$

where $\mathbf{v}_{w^t_{i+1}}$ denotes the row of $\mathbf{V}$ corresponding to the word of $w^t_{i+1}$. An analogous computation is performed for the previous sentence $s_{i-1}$.

**Objective.** Given a tuple $(s_{i-1}, s_i, s_{i+1})$, the objective optimized is the sum of the log-probabilities for the forward and backward sentences conditioned on the encoder representation:

$$
\sum_t \log P(w^t_{i+1} | w^{<t}_{i+1}, \mathbf{h}_i) + \sum_t \log P(w^t_{i-1} | w^{<t}_{i-1}, \mathbf{h}_i) \tag{10}
$$

The total objective is the above summed over all such training tuples.

## 2.2 Vocabulary expansion

We now describe how to expand our encoder's vocabulary to words it has not seen during training. Suppose we have a model that was trained to induce word representations, such as word2vec. Let $\mathcal{V}_{w2v}$ denote the word embedding space of these word representations and let $\mathcal{V}_{rnn}$ denote the RNN word embedding space. We assume the vocabulary of $\mathcal{V}_{w2v}$ is much larger than that of $\mathcal{V}_{rnn}$. Our goal is to construct a mapping $f : \mathcal{V}_{w2v} \rightarrow \mathcal{V}_{rnn}$ parameterized by a matrix $\mathbf{W}$ such that $\mathbf{v}' = \mathbf{W}\mathbf{v}$ for $\mathbf{v} \in \mathcal{V}_{w2v}$ and $\mathbf{v}' \in \mathcal{V}_{rnn}$. Inspired by [15], which learned linear mappings between translation word spaces, we solve an un-regularized L2 linear regression loss for the matrix $\mathbf{W}$. Thus, any word from $\mathcal{V}_{w2v}$ can now be mapped into $\mathcal{V}_{rnn}$ for encoding sentences.

## 3 Experiments

In our experiments, we evaluate the capability of our encoder as a generic feature extractor after training on the BookCorpus dataset. Our experimentation setup on each task is as follows:

- Using the learned encoder as a feature extractor, extract skip-thought vectors for all sentences.
- If the task involves computing scores between pairs of sentences, compute component-wise features between pairs. This is described in more detail specifically for each experiment.
- Train a *linear* classifier on top of the extracted features, with no additional fine-tuning or back-propagation through the skip-thoughts model.

We restrict ourselves to linear classifiers for two reasons. The first is to directly evaluate the representation quality of the computed vectors. It is possible that additional performance gains can be made throughout our experiments with non-linear models but this falls out of scope of our goal. Furthermore, it allows us to better analyze the strengths and weaknesses of the learned representations. The second reason is that reproducibility now becomes very straightforward.

## 3.1 Details of training

To induce skip-thought vectors, we train two separate models on our book corpus. One is a unidirectional encoder with 2400 dimensions, which we subsequently refer to as **uni-skip**. The other is a bidirectional model with forward and backward encoders of 1200 dimensions each. This model contains two encoders with different parameters: one encoder is given the sentence in correct order, while the other is given the sentence in reverse. The outputs are then concatenated to form a 2400 dimensional vector. We refer to this model as **bi-skip**. For training, we initialize all recurrent matricies with orthogonal initialization [16]. Non-recurrent weights are initialized from a uniform distribution in [-0.1,0.1]. Mini-batches of size 128 are used and gradients are clipped if the norm of the parameter vector exceeds 10. We used the Adam algorithm [17] for optimization. Both models were trained for roughly two weeks. As an additional experiment, we also report experimental results using a combined model, consisting of the concatenation of the vectors from uni-skip and bi-skip, resulting in a 4800 dimensional vector. We refer to this model throughout as **combine-skip**.

After our models are trained, we then employ vocabulary expansion to map word embeddings into the RNN encoder space. The publically available CBOW word vectors are used for this purpose [2]. The skip-thought models are trained with a vocabulary size of 20,000 words. After removing multiple word examples from the CBOW model, this results in a vocabulary size of 930,911 words. Thus even though our skip-thoughts model was trained with only 20,000 words, after vocabulary expansion we can now successfully encode 930,911 possible words.

Since our goal is to evaluate skip-thoughts as a general feature extractor, we keep text pre-processing to a minimum. When encoding new sentences, no additional preprocessing is done other than basic tokenization. This is done to test the robustness of our vectors. As an additional baseline, we also consider the mean of the word vectors learned from the uni-skip model. We refer to this baseline as **bow**. This is to determine the effectiveness of a standard baseline trained on the BookCorpus.

## 3.2 Semantic relatedness

Our first experiment is on the SemEval 2014 Task 1: semantic relatedness SICK dataset [30]. Given two sentences, our goal is to produce a score of how semantically related these sentences are, based on human generated scores. Each score is the average of 10 different human annotators. Scores take values between 1 and 5. A score of 1 indicates that the sentence pair is not at all related, while

| Method | $r$ | $\rho$ | MSE |
|---|---|---|---|
| Illinois-LH [18] | 0.7993 | 0.7538 | 0.3692 |
| UNAL-NLP [19] | 0.8070 | 0.7489 | 0.3550 |
| Meaning Factory [20] | 0.8268 | 0.7721 | 0.3224 |
| ECNU [21] | 0.8414 | – | – |
| Mean vectors [22] | 0.7577 | 0.6738 | 0.4557 |
| DT-RNN [23] | 0.7923 | 0.7319 | 0.3822 |
| SDT-RNN [23] | 0.7900 | 0.7304 | 0.3848 |
| LSTM [22] | 0.8528 | 0.7911 | 0.2831 |
| Bidirectional LSTM [22] | 0.8567 | 0.7966 | 0.2736 |
| Dependency Tree-LSTM [22] | **0.8676** | **0.8083** | **0.2532** |
| bow | 0.7823 | 0.7235 | 0.3975 |
| uni-skip | 0.8477 | 0.7780 | 0.2872 |
| bi-skip | 0.8405 | 0.7696 | 0.2995 |
| combine-skip | 0.8584 | 0.7916 | 0.2687 |
| combine-skip+COCO | 0.8655 | 0.7995 | 0.2561 |

| Method | Acc | F1 |
|---|---|---|
| feats [24] | 73.2 | |
| RAE+DP [24] | 72.6 | |
| RAE+feats [24] | 74.2 | |
| RAE+DP+feats [24] | 76.8 | 83.6 |
| FHS [25] | 75.0 | 82.7 |
| PE [26] | 76.1 | 82.7 |
| WDDP [27] | 75.6 | 83.0 |
| MTMETRICS [28] | 77.4 | 84.1 |
| TF-KLD [29] | **80.4** | **86.0** |
| bow | 67.8 | 80.3 |
| uni-skip | 73.0 | 81.9 |
| bi-skip | 71.2 | 81.2 |
| combine-skip | 73.0 | 82.0 |
| combine-skip + feats | 75.8 | 83.0 |

Table 3: **Left:** Test set results on the SICK semantic relatedness subtask. The evaluation metrics are Pearson's $r$, Spearman's $\rho$, and mean squared error. The first group of results are SemEval 2014 submissions, while the second group are results reported by [22]. **Right:** Test set results on the Microsoft Paraphrase Corpus. The evaluation metrics are classification accuracy and F1 score. Top: recursive autoencoder variants. Middle: the best published results on this dataset.

a score of 5 indicates they are highly related. The dataset comes with a predefined split of 4500 training pairs, 500 development pairs and 4927 testing pairs. All sentences are derived from existing image and video annotation datasets. The evaluation metrics are Pearson's $r$, Spearman's $\rho$, and mean squared error.

Given the difficulty of this task, many existing systems employ a large amount of feature engineering and additional resources. Thus, we test how well our learned representations fair against heavily engineered pipelines. Recently, [22] showed that learning representations with LSTM or Tree-LSTM for the task at hand is able to outperform these existing systems. We take this one step further and see how well our vectors learned from a completely different task are able to capture semantic relatedness when only a linear model is used on top to predict scores.

To represent a sentence pair, we use two features. Given two skip-thought vectors $u$ and $v$, we compute their component-wise product $u \cdot v$ and their absolute difference $|u - v|$ and concatenate them together. These two features were also used by [22]. To predict a score, we use the same setup as [22]. Let $r^\top = [1, \ldots, 5]$ be an integer vector from 1 to 5. We compute a distribution $p$ as a function of prediction scores $y$ given by $p_i = y - \lfloor y \rfloor$ if $i = \lfloor y \rfloor + 1$, $p_i = \lfloor y \rfloor - y + 1$ if $i = \lfloor y \rfloor$ and 0 otherwise. These then become our targets for a logistic regression classifier. At test time, given new sentence pairs we first compute targets $\hat{p}$ and then compute the related score as $r^\top \hat{p}$. As an additional comparison, we also explored appending features derived from an image-sentence embedding model trained on COCO (see section 3.4). Given vectors $u$ and $v$, we obtain vectors $u'$ and $v'$ from the learned linear embedding model and compute features $u' \cdot v'$ and $|u' - v'|$. These are then concatenated to the existing features.

Table 3 (left) presents our results. First, we observe that our models are able to outperform all previous systems from the SemEval 2014 competition. It highlights that skip-thought vectors learn representations that are well suited for semantic relatedness. Our results are comparable to LSTMs whose representations are trained from scratch on this task. Only the dependency tree-LSTM of [22] performs better than our results. We note that the dependency tree-LSTM relies on parsers whose training data is very expensive to collect and does not exist for all languages. We also observe using features learned from an image-sentence embedding model on COCO gives an additional performance boost, resulting in a model that performs on par with the dependency tree-LSTM. To get a feel for the model outputs, Table 4 shows example cases of test set pairs. Our model is able to accurately predict relatedness on many challenging cases. On some examples, it fails to pick up on small distinctions that drastically change a sentence meaning, such as *tricks on a motorcycle* versus *tricking a person on a motorcycle*.

## 3.3 Paraphrase detection

The next task we consider is paraphrase detection on the Microsoft Research Paraphrase Corpus [31]. On this task, two sentences are given and one must predict whether or not they are

| Sentence 1 | Sentence 2 | GT | pred |
|---|---|---|---|
| A little girl is looking at a woman in costume | A young girl is looking at a woman in costume | 4.7 | 4.5 |
| A little girl is looking at a woman in costume | The little girl is looking at a man in costume | 3.8 | 4.0 |
| A little girl is looking at a woman in costume | A little girl in costume looks like a woman | 2.9 | 3.5 |
| A sea turtle is hunting for fish | A sea turtle is hunting for food | 4.5 | 4.5 |
| A sea turtle is not hunting for fish | A sea turtle is hunting for fish | 3.4 | 3.8 |
| A man is driving a car | The car is being driven by a man | 5 | 4.9 |
| There is no man driving the car | A man is driving a car | 3.6 | 3.5 |
| A large duck is flying over a rocky stream | A duck, which is large, is flying over a rocky stream | 4.8 | 4.9 |
| A large duck is flying over a rocky stream | A large stream is full of rocks, ducks and flies | 2.7 | 3.1 |
| A person is performing acrobatics on a motorcycle | A person is performing tricks on a motorcycle | 4.3 | 4.4 |
| A person is performing tricks on a motorcycle | The performer is tricking a person on a motorcycle | 2.6 | 4.4 |
| Someone is pouring ingredients into a pot | Someone is adding ingredients to a pot | 4.4 | 4.0 |
| Nobody is pouring ingredients into a pot | Someone is pouring ingredients into a pot | 3.5 | 4.2 |
| Someone is pouring ingredients into a pot | A man is removing vegetables from a pot | 2.4 | 3.6 |

Table 4: Example predictions from the SICK test set. GT is the ground truth relatedness, scored between 1 and 5. The last few results show examples where slight changes in sentence structure result in large changes in relatedness which our model was unable to score correctly.

| | COCO Retrieval | | | | | | | |
|---|---|---|---|---|---|---|---|---|
| | Image Annotation | | | | Image Search | | | |
| Model | R@1 | R@5 | R@10 | Med $r$ | R@1 | R@5 | R@10 | Med $r$ |
| Random Ranking | 0.1 | 0.6 | 1.1 | 631 | 0.1 | 0.5 | 1.0 | 500 |
| DVSA [32] | 38.4 | 69.6 | 80.5 | **1** | 27.4 | **60.2** | 74.8 | **3** |
| GMM+HGLMM [33] | 39.4 | 67.9 | 80.9 | 2 | 25.1 | 59.8 | 76.6 | 4 |
| m-RNN [34] | **41.0** | **73.0** | **83.5** | 2 | **29.0** | 42.2 | **77.0** | **3** |
| bow | 33.6 | 65.8 | 79.7 | 3 | 24.4 | 57.1 | 73.5 | 4 |
| uni-skip | 30.6 | 64.5 | 79.8 | 3 | 22.7 | 56.4 | 71.7 | 4 |
| bi-skip | 32.7 | 67.3 | 79.6 | 3 | 24.2 | 57.1 | 73.2 | 4 |
| combine-skip | 33.8 | 67.7 | 82.1 | 3 | 25.9 | 60.0 | 74.6 | 4 |

Table 5: COCO test-set results for image-sentence retrieval experiments. **R@K** is Recall@K (high is good). **Med** $r$ is the median rank (low is good).

paraphrases. The training set consists of 4076 sentence pairs (2753 which are positive) and the test set has 1725 pairs (1147 are positive). We compute a vector representing the pair of sentences in the same way as on the SICK dataset, using the component-wise product $u \cdot v$ and their absolute difference $|u - v|$ which are then concatenated together. We then train logistic regression on top to predict whether the sentences are paraphrases. Cross-validation is used for tuning the L2 penalty.

As in the semantic relatedness task, paraphrase detection has largely been dominated by extensive feature engineering, or a combination of feature engineering with semantic spaces. We report experiments in two settings: one using the features as above and the other incorporating basic statistics between sentence pairs, the same features used by [24]. These are referred to as *feats* in our results. We isolate the results and baselines used in [24] as well as the top published results on this task.

Table 3 (right) presents our results, from which we can observe the following: (1) skip-thoughts alone outperform recursive nets with dynamic pooling when no hand-crafted features are used, (2) when other features are used, recursive nets with dynamic pooling works better, and (3) when skip-thoughts are combined with basic pairwise statistics, it becomes competitive with the state-of-the-art which incorporate much more complicated features and hand-engineering. This is a promising result as many of the sentence pairs have very fine-grained details that signal if they are paraphrases.

## 3.4   Image-sentence ranking

We next consider the task of retrieving images and their sentence descriptions. For this experiment, we use the Microsoft COCO dataset [35] which is the largest publicly available dataset of images with high-quality sentence descriptions. Each image is annotated with 5 captions, each from different annotators. Following previous work, we consider two tasks: image annotation and image search. For image annotation, an image is presented and sentences are ranked based on how well they describe the query image. The image search task is the reverse: given a caption, we retrieve images that are a good fit to the query. The training set comes with over 80,000 images each with 5 captions. For development and testing we use the same splits as [32]. The development and test sets each contain 1000 images and 5000 captions. Evaluation is performed using Recall@K, namely the mean number of images for which the correct caption is ranked within the top-K retrieved results

(and vice-versa for sentences). We also report the median rank of the closest ground truth result from the ranked list.

The best performing results on image-sentence ranking have all used RNNs for encoding sentences, where the sentence representation is learned jointly. Recently, [33] showed that by using Fisher vectors for representing sentences, linear CCA can be applied to obtain performance that is as strong as using RNNs for this task. Thus the method of [33] is a strong baseline to compare our sentence representations with. For our experiments, we represent images using 4096-dimensional OxfordNet features from their 19-layer model [36]. For sentences, we simply extract skip-thought vectors for each caption. The training objective we use is a pairwise ranking loss that has been previously used by many other methods. The only difference is the scores are computed using only linear transformations of image and sentence inputs. The loss is given by:

$$\sum_{\mathbf{x}} \sum_{k} \max\{0, \alpha - s(\mathbf{Ux}, \mathbf{Vy}) + s(\mathbf{Ux}, \mathbf{Vy}_k)\} + \sum_{\mathbf{y}} \sum_{k} \max\{0, \alpha - s(\mathbf{Vy}, \mathbf{Ux}) + s(\mathbf{Vy}, \mathbf{Ux}_k)\},$$

where $\mathbf{x}$ is an image vector, $\mathbf{y}$ is the skip-thought vector for the groundtruth sentence, $\mathbf{y}_k$ are vectors for constrastive (incorrect) sentences and $s(\cdot, \cdot)$ is the image-sentence score. Cosine similarity is used for scoring. The model parameters are $\{\mathbf{U}, \mathbf{V}\}$ where $\mathbf{U}$ is the image embedding matrix and $\mathbf{V}$ is the sentence embedding matrix. In our experiments, we use a 1000 dimensional embedding, margin $\alpha = 0.2$ and $k = 50$ contrastive terms. We trained for 15 epochs and saved our model anytime the performance improved on the development set.

Table 5 illustrates our results on this task. Using skip-thought vectors for sentences, we get performance that is on par with both [32] and [33] except for R@1 on image annotation, where other methods perform much better. Our results indicate that skip-thought vectors are representative enough to capture image descriptions without having to learn their representations from scratch. Combined with the results of [33], it also highlights that simple, scalable embedding techniques perform very well provided that high-quality image and sentence vectors are available.

### 3.5 Classification benchmarks

For our final quantitative experiments, we report results on several classification benchmarks which are commonly used for evaluating sentence representation learning methods.

We use 5 datasets: movie review sentiment (MR), customer product reviews (CR), subjectivity/objectivity classification (SUBJ), opinion polarity (MPQA) and question-type classification (TREC). On all datasets, we simply extract skip-thought vectors and train a logistic regression classifier on top. 10-fold cross-validation is used for evaluation on the first 4 datasets, while TREC has a pre-defined train/test split. We tune the L2 penality using cross-validation (and thus use a nested cross-validation for the first 4 datasets).

| Method | MR | CR | SUBJ | MPQA | TREC |
|---|---|---|---|---|---|
| NB-SVM [37] | 79.4 | 81.8 | 93.2 | 86.3 | |
| MNB [37] | 79.0 | 80.0 | 93.6 | 86.3 | |
| cBoW [6] | 77.2 | 79.9 | 91.3 | 86.4 | 87.3 |
| GrConv [6] | 76.3 | 81.3 | 89.5 | 84.5 | 88.4 |
| RNN [6] | 77.2 | 82.3 | 93.7 | 90.1 | 90.2 |
| BRNN [6] | 82.3 | 82.6 | 94.2 | 90.3 | 91.0 |
| CNN [4] | 81.5 | 85.0 | 93.4 | 89.6 | **93.6** |
| AdaSent [6] | **83.1** | **86.3** | **95.5** | **93.3** | 92.4 |
| Paragraph-vector [7] | 74.8 | 78.1 | 90.5 | 74.2 | 91.8 |
| bow | 75.0 | 80.4 | 91.2 | 87.0 | 84.8 |
| uni-skip | 75.5 | 79.3 | 92.1 | 86.9 | 91.4 |
| bi-skip | 73.9 | 77.9 | 92.5 | 83.3 | 89.4 |
| combine-skip | 76.5 | 80.1 | 93.6 | 87.1 | 92.2 |
| combine-skip + NB | 80.4 | 81.3 | 93.6 | 87.5 | |

Table 6: Classification accuracies on several standard benchmarks. Results are grouped as follows: (a): bag-of-words models; (b): supervised compositional models; (c) Paragraph Vector (unsupervised learning of sentence representations); (d) ours. Best results overall are **bold** while best results outside of group (b) are underlined.

On these tasks, properly tuned bag-of-words models have been shown to perform exceptionally well. In particular, the NB-SVM of [37] is a fast and robust performer on these tasks. Skip-thought vectors potentially give an alternative to these baselines being just as fast and easy to use. For an additional comparison, we also see to what effect augmenting skip-thoughts with bi-gram Naive Bayes (NB) features improves performance [3].

Table 6 presents our results. On most tasks, skip-thoughts performs about as well as the bag-of-words baselines but fails to improve over methods whose sentence representations are learned directly for the task at hand. This indicates that for tasks like sentiment classification, tuning the representations, even on small datasets, are likely to perform better than learning a generic unsupervised

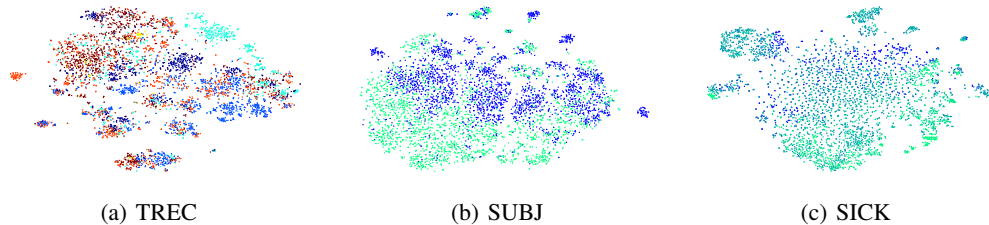

|            (a) TREC            |            (b) SUBJ            |            (c) SICK            |

Figure 2: t-SNE embeddings of skip-thought vectors on different datasets. Points are colored based on their labels (question type for TREC, subjectivity/objectivity for SUBJ). On the SICK dataset, each point represents a sentence pair and points are colored on a gradient based on their relatedness labels. Results best seen in electronic form.

sentence vector on much bigger datasets. Finally, we observe that the skip-thoughts-NB combination is effective, particularly on MR. This results in a very strong new baseline for text classification: combine skip-thoughts with bag-of-words and train a linear model.

### 3.6   Visualizing skip-thoughts

As a final experiment, we applied t-SNE [38] to skip-thought vectors extracted from TREC, SUBJ and SICK datasets and the visualizations are shown in Figure 2. For the SICK visualization, each point represents a sentence pair, computed using the concatenation of component-wise and absolute difference of features. Even without the use of relatedness labels, skip-thought vectors learn to accurately capture this property.

## 4   Conclusion

We evaluated the effectiveness of skip-thought vectors as an off-the-shelf sentence representation with linear classifiers across 8 tasks. Many of the methods we compare against were only evaluated on 1 task. The fact that skip-thought vectors perform well on all tasks considered highlight the robustness of our representations.

We believe our model for learning skip-thought vectors only scratches the surface of possible objectives. Many variations have yet to be explored, including (a) deep encoders and decoders, (b) larger context windows, (c) encoding and decoding paragraphs, (d) other encoders, such as convnets. It is likely the case that more exploration of this space will result in even higher quality representations.

### Acknowledgments

We thank Geoffrey Hinton for suggesting the name skip-thoughts. We also thank Felix Hill, Kelvin Xu, Kyunghyun Cho and Ilya Sutskever for valuable comments and discussion. This work was supported by NSERC, Samsung, CIFAR, Google and ONR Grant N00014-14-1-0232.

## Footnotes

[1]A preliminary version of our model was developed in the context of a computer vision application [9].

[2]http://code.google.com/p/word2vec/

[3]We use the code available at `https://github.com/mesnilgr/nbsvm`

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
