[Reviews · NeurIPS 2015]

Submitted by Assigned_Reviewer_1

To clarify after reading author response: the pre-trained sentence vectors are going to be easy to use and thus will be most certainly useful. What I meant was that to retrain the model would not be as easy as using, say, word2vec.

The package shared on GitHub does not seem to include the training part yet.
Summary: The paper extends an interesting application of the sequence to sequence models by borrowing the idea of skip-gram, using the current sentence to predict the sentence before and after. Although the experiments do show the potential of this model, the approach didn't really outperform the state-of-the-art on any of the tested cases. Given that the model takes a lot of time to train, it might be difficult for users to retrain the vectors using their, domain-specific, corpus.

Submitted by Assigned_Reviewer_2

The authors propose an encoder-decoder approach to learning vector representations of entire sentences. In much the same way that CBOW exploits the distributional semantics of words, the hope is that in natural discourse, sentences will obey similar distributional properties. If the hypothesis holds then training a sentence to predict its neighboring sentences would produce good learned representations of sentences. This approach is particularly appealing, because like CBOW/SkipGram, the model can be trained in an unsupervised fashion using the data itself to constrain the weights.

They also propose and study a particular embodiment of this idea in which they employ gated RNNs for encoding and decoding functions. As noted by the authors, similar models have been used for machine translation, and you can think of this model as "translating" a current sentence into a previous and next sentence. The authors evaluate the model on eight different tasks, showing the learned embeddings (the hidden states of the encoder) are semantically meaningful and useful for training classifiers (competitive or better than many existing systems).

Overall, the paper is well written, but I have a few comments:

*In the experimental results section, it's not always clear if the number reported for a system is from the literature or if it's reproduced from scratch. For example, in the first results table (Table 3), it's clear that the numbers from the challenge are ones reported in the literature, but in Table 6 it's not clear if paragraph vector was re-trained on the same data as the skip-vector model (the book data), or if the reported number is from the literature.

*Another potential weakness of the particular gated RNN implementation of skip-thoughts is that compared to paragraph vectors, it is much more difficult to train (more than a week to train?). While I can see that paragraph vector requires inference to embed new sentences, is this inference really that much more expensive than having the gated RNN consume the sentence one token at a time?

*It might be worth extending Table4 with additional sentence pairs that show the cases in which skip-thoughts improve performance over the baseline. A proxy for this would be to search for sentences with poor lexical similarity, but high semantic similarity (both predicted and true semantic similarity).
Summary: A well written paper with a novel contribution and thorough experimental evaluation against numerous baselines on eight different tasks.

Submitted by Assigned_Reviewer_3

I think the idea of extending distributional methods from words to sentences is great.

The experiments look carefully done and they compare on a large variety of different problems.

Summary: This paper introduces an unsupervised way to train sentence vector representations. The idea is to take GRU to encode a sentence and then predict words of surrounding sentences.

I like this paper a lot. I would strongly encourage to change it to skip-sentence vector though, sentence /= thought ;)

Submitted by Assigned_Reviewer_4

The authors present a technique inspired by the Skip-gram model of Tomas Mikolov to encode a sentence representation. The representation is the last layer in the sentence sequence of a GRU RNN. From this representation, the model is trained on a Billion words corpus (extracted from books) to generate the previous and next sentence in a paragraph as a GRU RNN Language Model conditioned on the encoded representation. Even if the model is trained on a relatively small vocabulary (20K), they present an easy way to extend it to 1M using word2vec and a linear mapping. They present an extensive set of experiments across many tasks for NLP, using the raw sentence representation as feature.

I really liked the paper and I think it is a good starting point for upcoming research in learning sentence representation, potentially learning representation at the paragraph level. Potential criticism of the paper would be that there is no new state of the art from the method but the research community knows how hard it is to find a representation that generalizes well across a wide range of NLP tasks.

The only experiment missing in my opinion would be to fine-tune through the GRU RNN from the sentence representation on a few datasets described in the paper to verify how far the obtained improvement would be from the state of the art. Regarding the bidirectional model, do you have results with a single model that would take the forward and backward pass rather than concatenating the representation?

minor comments: 2nd 3rd paragraph in Introduction - Fig 2 and Table 2 should be replaced by Fig 1 and Table 1 Sec 1 last paragraph 'such an *experimental* is begin' Sec 3.2 2nd paragraph 'learned representation *fair* against heavily'

training details: how many words per seconds the model is able to process for training using adam? how many passes thourgh the whole dataset do you perform in the 2 weeks training?
Summary: I really liked the paper and I think it is a good starting point for upcoming research in learning sentence representation, potentially learning representation at the paragraph level.

Submitted by Assigned_Reviewer_5

TL; DR This paper presents a natural generalization of neural word embeddings: sentence embeddings! The training objective of the proposed "skip-thoughts" model is to predict the previous and next sentence given an encoding of the current sentence. Thus, it may be trained without supervision from raw text, and the resulting embeddings used as "features" for downstream tasks. The authors present experiments on 8 tasks, and while the results of the proposed method are not state-of-the-art, it outperforms many previous baselines with less feature engineering.

The proposed "skip-thoughts" model is a fairly ingenious application of the recently popularized sequence-to-sequence RNN framework. It's been known for a long time now (e.g., Ando & Zhang, 2005) that learning representations of observed data X via auxiliary tasks can be helpful in downstream tasks, in which one is interested in making predictions Y. However, doing so at the sentence level is a nice twist, though not an entirely novel one (e.g. the cited paragraph vector work, and earlier work cited therein). I think the technical approach in the paper, being largely derived from existing RNN frameworks, is sensible. I also liked the vocabulary expansion strategy.

I do have some concerns regarding the experimental evaluation. The main question in my mind is if the proposed approach is a better way of learning from a big pile of unlabeled data X than alternatives, in terms of downstream performance on predicting Y. Unfortunately, I don't believe this question is addressed in the experiments; there is no baseline which uses to the same X (the Book11K corpus). Thus, it is not clear if the reported results are simply as a result of (a) using a lot more data, or (b) the model is capturing something interesting beyond superficial word co-occurence statistics.

I would have liked to see a comparison to a strong baseline which uses the same training data. Short of this, it would have at least been good to see a *weak* baseline, e.g. a bag-of-words model. Yet another option would have been to train word embeddings on Book11K, and use these in some of the more engineered baselines so that at least the data conditions are matched.

One common issue with RNNs in the encoder-decoder framework is the encoder must summarize the entire input in a single vector. One heuristic that has been used to alleviate this problem is to reverse the order of the tokens. More recently, attention models have been used. I would have liked to see a little discussion of this issue. In a similar vein, I would have liked to see more details about:

-how the proposed approach was tuned, and how sensitive it is to hyperparameters

-what alternate architectures were used, if any. For instance, an obvious alternative is to only predict the next sentence given the current sentence (it's not 100% obvious why it makes sense to predict sentences in reverse, even though I'm willing to believe it's helpful as it doubles the amount of data).

-how performance improves with increasing amounts of unlabeled data X (presumably could be derived from model checkpoints from a prior training run).

UPDATE AFTER AUTHOR RESPONSE:

Including a baseline that uses word-level embeddings trained on Book11K would improve the paper. However, it seems to me that the natural baseline to compare to is the paragraph vector model [1], with matched train and test conditions. I don't understand why this comparison wasn't made; perhaps this should be clarified in the paper.

[1] http://arxiv.org/pdf/1405.4053v2.pdf

Summary: The proposed "skip-thoughts" model is a fairly ingenious application of the recently popularized sequence-to-sequence RNN framework. However, I worry that the experimental evaluation focuses on conditions in which there is a data mismatch between the baselines and the proposed approach.

Author Feedback
Author rebuttal: We thank the reviewers for their valuable feedback.

R1:

All of the experimental results that are not from our models were reported in other papers. We will make this clear.

Re table 4: that is a good idea. We realized that the prediction scores of the competition teams are publicly available. We can add further analysis between methods in the supplementary material.

R2:

Bidirectional variant: we didn't try this but it is definitely worth trying out.

We were able to do roughly 1 pass through the dataset during that time. Each weight update for a minibatch took roughly 1-3 seconds depending on the sentence length.

R3:

The potential mismatch between dataset and method is a fair point. However, it is very unlikely that a simple method (e.g bag-of-words) whose vectors were trained on Book11K would do well on downstream tasks. The publicly available word2vec embeddings were trained on a much larger dataset (100 billion words, compared to Book11K which was roughly 1 billion words). These embeddings are often used as baselines for the same and related tasks, often just averaging the vectors. In almost all cases, these results perform rather poorly, e.g:

- "DeViSE" baseline from "Deep Fragment Embeddings for Bidirectional Image Sentence Mapping" (Karpathy et al, 2014)
- "SG" from table 3 of "Sparse Overcomplete Word Vector Representations" (Faruqui et al, 2015), 77.8% on TREC compared to 91% with skip-thoughts

That being said, we are happy to include an additional baseline for each experiment using the RNN word embeddings that were learned on Book11K.

Regarding other architectures, we believe there is a whole family of models that would do well on the same experimental setup, e.g. using different encoders, predicting more than one sentence before and after, etc. The goal of our paper was to demonstrate the one can learn generic sentence vectors that work well on many highly diverse tasks - something that has not been shown before with neural networks. A large comparison of other architectures would be very interesting but would likely require a paper on its own. In some sense, our result is largely a proof of concept that this kind of setup works using distributed sentence representations.

The only hyperparameter we tuned was the size of the hidden state. The bigger the hidden state was, the better the results got. We used 2400 since it was the biggest we could comfortably fit onto the GPU. All other hyperparameters were set to reasonable defaults that generally work well.

Regarding performance improvements: we did this, periodically checking the validation results on each of the tasks as the model was training. The results kept improving but started to saturate after about a week.

R4:

"Vocabulary expansion relies on pre-trained word vectors and seems ad-hoc"
What exactly is ad-hoc about it? It builds on strong and fast word level methods which can encode large vocabularies. It is a simple and clean approach that requires nothing more than 1 run of linear regression on public word vectors. Consequently, one can get a vocabulary of 1 million words for essentially free.

"Experimental results rely too much on vocabulary expansion (original vocabulary is too small)"
We respectfully disagree. The model works fine without vocabulary expansion, just a bit worse. Unless of course the dataset has a widely different vocabulary than books. (e.g. on movie reviews, one can get about 73% accuracy without vocabulary expansion, compared to about 75% with it) Furthermore, once the regression weights are learned the word vectors can be cached and so one does not incur any additional cost in using it.

"Not clear how to address words that are rare in the training set"
This is exactly what vocabulary expansion solves.

R5:

"Given that the model takes a lot of time to train, the empirical value of this model remains unclear."
Our code is already publicly available on GitHub to reproduce each experiment in the paper and has already received over 200 stars. Thus, the empirical value of the model clearly exists.